# Outer hair cells stir cochlear fluids

**Choongheon Lee[1,2†], Mohammad Shokrian[2†], Kenneth S Henry[1,3,4†], Laurel H Carney[3,4,5], J Christopher Holt[1,4], Jong-Hoon Nam[2,3,4]***

[1]Department of Otolaryngology, University of Rochester, Rochester, United States; [2]Department of Mechanical Engineering, University of Rochester, Rochester, United States; [3]Department of Biomedical Engineering, University of Rochester, Rochester, United States; [4]Department of Neuroscience, University of Rochester, Rochester, United States; [5]Department of Electrical and Computer Engineering, University of Rochester, Rochester, United States

## eLife Assessment

Although others have proposed that OHC electromotility subserves cochlear amplification by acting as a 'fluid pump', and evidence for this has been found using electrical stimulation of excised cochleae, this **fundamental** study substantially advances our understanding of cochlear homeostasis. This is the first report to test the pumping effect in vivo and consider its implications for cochlear homeostasis and drug delivery. The manuscript provides **compelling** evidence for OHC-based fluid flow within the cochlea.

**\*For correspondence:**
jong-hoon.nam@rochester.edu

[†]These authors contributed equally to this work

**Competing interest:** The authors declare that no competing interests exist.

**Abstract** We hypothesized that active outer hair cells drive cochlear fluid circulation. The hypothesis was tested by delivering the neurotoxin, kainic acid, to the intact round window of young gerbil cochleae while monitoring auditory responses in the cochlear nucleus. Sounds presented at a modest level significantly expedited kainic acid delivery. When outer-hair-cell motility was suppressed by salicylate, the facilitation effect was compromised. A low-frequency tone was more effective than broadband noise, especially for drug delivery to apical locations. Computational model simulations provided the physical basis for our observation, which incorporated solute diffusion, fluid advection, fluid–structure interaction, and outer-hair-cell motility. Active outer hair cells deformed the organ of Corti like a peristaltic tube to generate apically streaming flows along the tunnel of Corti and basally streaming flows along the scala tympani. Our measurements and simulations coherently suggest that active outer hair cells in the tail region of cochlear traveling waves drive cochlear fluid circulation.

## Introduction

The inner-ear labyrinth is filled with two lymphatic fluids, the perilymph and endolymph. An electrochemical potential difference between the two lymphatic fluids drives hair-cell mechano-transduction (*Fettiplace and Kim, 2014*). Although the inner-ear fluid system is connected to the cerebrospinal fluid of the brain through the cochlear and vestibular aqueducts and to the vascular system across the blood–labyrinth barrier, inner-ear fluid is virtually isolated from other body-fluid systems (*Salt et al., 1986*; *Ohyama et al., 1988*). Consequently, intervening in hearing health by delivering substances to the inner-ear fluid through the systemic circulation is challenging. The maintenance of inner-ear fluid composition is achieved locally by the secretion and absorption of ions and molecules through inner-ear epithelial cells (*Wangemann and Marcus, 2017*). As inner-ear fluid homeostasis is maintained locally, longitudinal electro-chemical gradients, including the endocochlear potential, may vary

along the cochlear length (*Schulte and Schmiedt, 1992*; *Sadanaga and Morimitsu, 1995*; *Hirose and Liberman, 2003*).

The inner-ear fluid has been reported to be stationary, unlike other lymphatic fluids in the body (*Salt and Hirose, 2018a*). Thus, it is considered that substance transport in the inner-ear fluid is governed by diffusion (*Salt and Plontke, 2005*). Diffusion is an effective mechanism for a substance to travel along sub-micrometer distances (*Berg, 1993*). For instance, considering the diffusion speed of small molecules in water, it takes microseconds for neurotransmitters to diffuse across a 20 nm synaptic gap. In contrast, diffusion is inefficient for travel on the centimeter scale. It takes days for a drug applied at the round window to travel 30 mm to the apical end of the human cochlea. In practice, the substance would not reach the apex because it would be resorbed before traveling this distance. Studies on inner-ear drug delivery confirm this principle of physics: longitudinal mass transport along the inner-ear labyrinth is challenging (*Salt et al., 2016*; *Lukashkin et al., 2020*).

The two membranous openings of the inner ear, the oval and round windows, are sites for noninvasive local delivery of substances into the inner ear. Intratympanic injections of antibiotics and anti-inflammatory drugs have been attempted to manage inner-ear disorders despite limited efficacy (*Lavigne et al., 2016*; *Salt and Plontke, 2018b*). Releasing a substance at the end of a long, closed tube forms a steep gradient of concentration. Consequently, the substance does not reach distant regions (*Salt and Hirose, 2018a*), whereas, in the vicinity of drug release, the concentration is much higher than desired. One remedy is to inject fluid by making two perforations in the inner ear—one for intruding and the other for extruding (*Borkholder et al., 2014*; *Tandon et al., 2016*; *Talaei et al., 2019*). Such an invasive approach is often associated with the injection of a substantial fluid volume, larger than the entire perilymph in the inner ear (*Szeto et al., 2020*).

The tube-shaped organ of Corti (OoC) is lined with motile outer hair cells that are activated systematically with a large phase velocity (greater than a few m/s) toward the apex (*Olson et al., 2012*). A previous study suggests that outer-hair-cell motility can cause oscillatory fluid motion along the triangular fluid channel within the OoC (*Karavitaki and Mountain, 2007*). Recent measurements of OoC vibrations provide a possible explanation for how this longitudinal fluid flow is generated: outer hair cells operate like an area motor by deforming the OoC cross-sectional area (*Jabeen et al., 2020*; *Guinan, 2022*; *Lin et al., 2024*). Outer hair cells boost the OoC vibrations over the entire span of the traveling waves (*Lee et al., 2016*; *Cooper et al., 2018*; *He et al., 2018*; *Fallah et al., 2019*). In theory, peristaltic deformation of the OoC could induce non-oscillatory streaming along the tunnel of Corti (*Shokrian et al., 2020*). This peristaltic motion of the OoC has not been explored experimentally.

We hypothesized that outer hair cells agitated by sounds could induce advection of the cochlear fluids. To test this hypothesis, the transport of a neurotoxin along the scala tympani was measured. Kainic acid irreversibly damages afferent synapses with inner hair cells (*Sun et al., 2001*). Importantly, kainic acid does not affect the hair cells, unlike other popular substances used for inner-ear drug-delivery experiments, such as streptomycin and salicylate (*Salt et al., 2016*; *Lukashkin et al., 2020*). Kainic acid (10 mM) was administered at the round-window niche in young gerbils. Multichannel electrodes were inserted into the anteroventral cochlear nucleus (AVCN) to monitor drug effect along the tonotopic axis. The relation between the drug's effect time and location was obtained while controlling sound and activity of outer hair cells, two factors that have not been considered in previous drug-delivery studies. To provide theoretical support for experimental observations, computer models were used to simulate inner-ear drug delivery in three steps: the virtual cochlea simulated OoC vibrations due to sound (*Zhou and Nam, 2019*) the Navier–Stokes equation including a nonlinear convective term was solved to obtain the drift of the Corti and scala fluids (*Shokrian et al., 2020*) and the diffusion-advection equation was solved to simulate the spatiotemporal change of kainic acid concentration in the cochlear fluids.

## Results

Outer-hair-cell motility and sound properties, such as frequency, level, and bandwidth, were the parameters that we hypothesized to affect drug delivery. Testing these quantities and their combinations requires an extensive series of experiments. In this study, we present a control case and four test cases to focus on two major questions: whether sounds and outer-hair-cell motility affect inner-ear drug delivery. The control experiment was the drug delivery in near silence ('Silence'). The test cases included (1) broadband sounds (0.1–12 kHz, 60 or 75 dB sound pressure level [SPL]) presented during

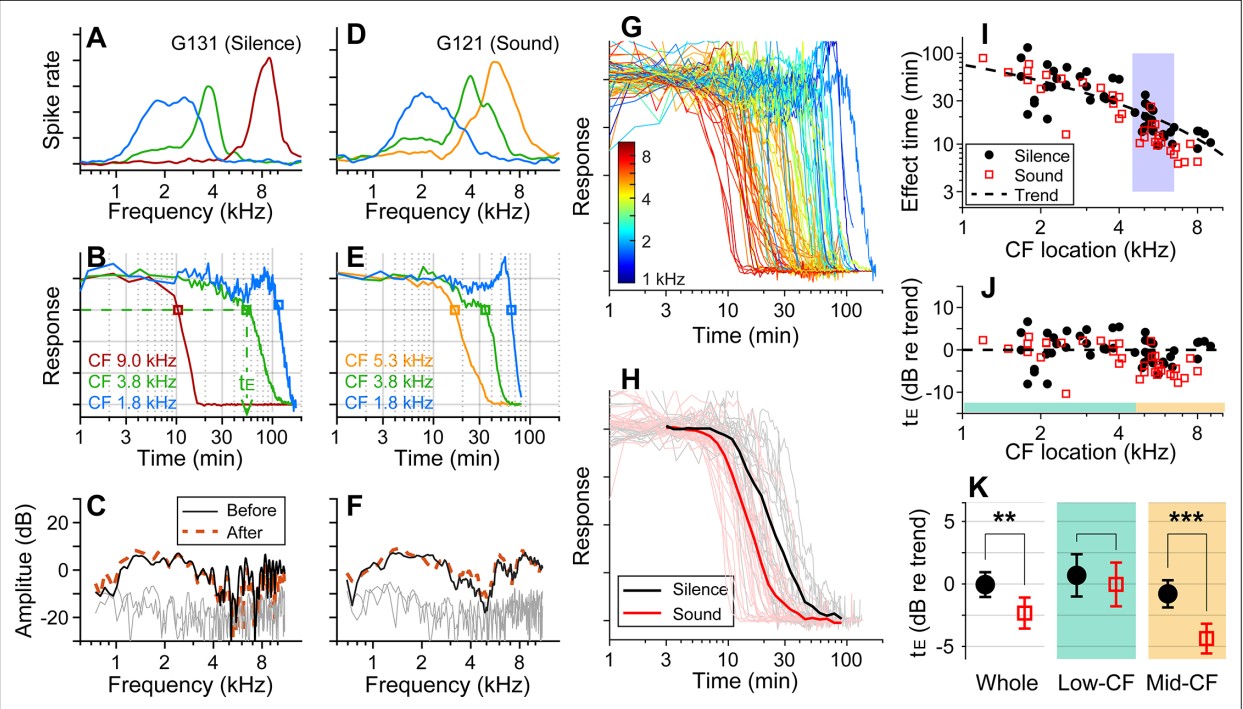

**Figure 1.** Sound-facilitated inner-ear drug delivery. Kainic acid was applied to the round window of young gerbils under different acoustic conditions—in silence or with 60–75 dB sound pressure level (SPL) sounds. (**A**) Tuning curves at 30 dB SPL. The characteristic frequencies (CFs) of the three probing channels are 9.0, 3.8, and 1.8 kHz. (**B**) Normalized neural responses in the anteroventral cochlear nucleus (AVCN) versus the time of drug application. The drug effect time, $t_E$, was defined at the 75 percentiles of normalized curves. In this measurement, it took 54 min to affect the 3.8 kHz CF location (green dashed line and arrow). (**C**) Distortion product otoacoustic emissions (DPOAEs) were measured before and after the kainic acid delivery. Two stimulating tones were at 40 dB SPL, and the frequency ratio was 1.1. The x-axis is the $f_1$ frequency. (**A–C**) Example of drug delivered in silence ('Silence' case). (**D–F**) Example of drug delivery during the presentation of broadband noise at 75 dB SPL ('Sound' case). (**G**) Response curves with colors representing CFs. Basal (higher frequency) responses decay earlier. Out of 82 measurements, 48 silence-case curves are from 18 animals and 34 sound-case curves are from 13 animals. (**H**) Mean responses of sound and silence cases at similar locations (CF = 4.5–6.5 kHz); n = 19 for Silence, and n = 13 for Sound. (**I**) Effect time versus CF location. The shaded frequency range corresponds to the data in (**H**). The broken curve ('Trend' line) was obtained by fitting a curve to the silence data using 1D diffusion theory. (**J**) Effect time in dB with respect to the trend line. $t_E$ in dB = $20\log_{10}(t_E/t_{Trend})$. (**K**) Two-tailed *t*-tests between the effect times of sound and silence cases for the entire CF range (whole, n = 48, 34 for silence and sound), low-CF locations (<4.5 kHz, n = 24, 16 for silence and sound), and high-CF locations (>4.5 kHz, n = 24, 18 for silence and sound). Throughout this article, the symbols and range bars indicate the mean and the 95% CI, respectively. When an individual data set was presented as an example, the subject identities are indicated as G###, where ### is a three-digit number.

drug delivery ('Sound'), (2) the same broadband sounds presented, but with salicylic acid administered before kainic acid application ('post-SA'), or an 80 dB SPL pure-tone at either (3) 0.5 kHz ('LF-tone'), or (4) 3–6 kHz ('MF-tone').

## Sounds affect inner-ear drug delivery

Before applying 10 mM kainic acid to the intact round window, the characteristic frequencies (CFs) of probing locations were identified (*Figure 1A and D*). Distortion product otoacoustic emissions (DPOAEs) were measured before and after drug delivery to confirm no change in outer-hair-cell motility during the experiment (*Figure 1C and F*). When pre- and post-delivery DPOAEs changed more than 5 dB at any frequency between 0.1 and 10 kHz, the results were excluded (four animals). In one set of experiments, over 90% of stimulus intervals were kept silent (approximately 55 s per minute, *Figure 1A–C*). In the other experiments, broadband sounds at a modest level (0.1–12 kHz, 60–75 dB SPL) were played during over 90% of the stimulus intervals (*Figure 1D–F*). Approximately 8% of time (4.6 s per minute) were used to acquire background neural activity and to measure responses to 30–40 dB SPL probe tones at three CFs. Under our experimental condition (4 µL of 10 mM kainic acid applied at the intact round window), neural responses were abolished after a few to several hundred minutes (*Figure 1B and E*). The effect time was defined as the latency following exposure at which

the neural response dropped below 75% of its baseline value ($t_E$, *Figure 1B*). The drug effect was tonotopic, with higher CF locations being affected earlier (*Figure 1G*). In *Figure 1H*, the sound- and the silence-case response curves from similar CF locations (between 4.5 and 6.5 kHz) were averaged for comparison. *Figure 1I* presents the drug delivery data as the effect time ($t_E$) versus CF location. We acquired 34 data points from 13 gerbils for the sound protocol and 48 data points from 18 animals for the silence protocol. The trend line (broken curve) was obtained by fitting a curve to the silence-case data points with the 1D diffusion equation (*Berg, 1993*).

To compare the effect times of the sound and silence cases despite varying delivery distances, the effect time was normalized by the trend values at respective CF locations and represented in dB ($t_E$ in dB = $20\log_{10}(t_E/t_{Trend})$, *Figure 1J*). Average $t_E$ over the entire CF range was significantly faster for the sound protocol than for the silent case (p=0.0045). Subsequent analyses of low- (<4.5 kHz) and mid-frequency (>4.5 kHz) regions further showed that broadband sound exposure increased the speed of drug delivery in the mid-CF region (>4.5 kHz, p<0.001) but not the low-CF region (<4.5 kHz, = 0.55; *Figure 1K*).

## Outer-hair-cell motility was required for the sound effect

Motile outer hair cells boost cochlear responses to sounds. If the sound effect on cochlear drug delivery depends on fluid agitation from outer-hair-cell motility, inhibiting the motility should produce effect times similar to the silent condition. To test this hypothesis, before delivering kainic acid, salicylic acid was administered intraperitoneally (200 mg/kg) to inhibit outer-hair-cell motility (*Figure 2A*). DPOAEs were recorded before and after the salicylic acid application to confirm its effect (*Figure 2B and E*). After salicylate injection, neural responses decayed for 20–30 min to reach a steady state (*Figure 2C and F*). Because the salicylate effect raised the hearing threshold, the probe-tone level was increased by 15–20 dB prior to kainic acid application (typically from 30 to 50 dB SPL; the discontinuity in the response curve near 70 min was due to this increase in the probe tone level). The red-dashed vertical lines of *Figure 2C and F* indicate the timing of kainic acid application. *Figure 2D and G* present only the kainic acid application responses (the time span after the dashed vertical lines in panels B and E). Note that 75 dB SPL broadband sounds were played in the post-SA protocol. The two examples in *Figure 2B–G* also demonstrate the variance of the salicylate effect. In one case, DPOAEs decreased by <10 dB (re: 40 dB stimulation) and the neural responses to 30 dB SPL sound decreased by 40% (*Figure 2C*). In the other case, DPOAEs decreased by approximately 20 dB, and the neural responses to 30 dB SPL sound decreased to the noise floor (*Figure 2F*). For the post-SA protocol, we acquired 24 data points from eight animals. The post-kainate response decay of salicylate-treated ears followed the trend of the silence case despite use of the sound protocol (*Figure 2H*). Despite the sound stimulation, the effect time of the post-SA case was not different from the silence case (p=0.096) and was greater than the sound case (p<0.001, *Figure 2J*). Like the silence versus sound protocols comparison, the statistical difference was from the mid-CF locations, not from low-CF locations (*Figure 2J*). The post-SA results support the hypothesis that outer-hair-cell motility is required for faster inner-ear drug delivery with sound exposure.

## Low-frequency tones were more effective

If the sound effect on cochlear drug delivery were due to fluid agitation, other sounds might be more effective than random noise. Vibrations along the cochlear partition propagate with decaying phase velocity toward the apex, before stalling shortly after reaching the sound's CF location. The advective flow of the cochlear fluid is shaped by the vibration frequency and amplitude and by the phase velocity of traveling waves (*Lighthill, 1992*; *Shokrian et al., 2020*). To test the hypothesis that the type of sound stimulus impacts the effectiveness of drug delivery, especially in low-frequency regions, pure-tones at 80 dB SPL were played during inner-ear drug delivery, instead of broadband noise. In choosing the frequency of the pure tone, we considered the nature of cochlear traveling waves, guided by fluid-dynamics simulations (*Zhou and Nam, 2019*; *Shokrian et al., 2020*). The frequency of 0.5 kHz was chosen close to the low bound of the gerbil's audible frequency range. The mid-frequency was selected from between 3 and 6 kHz to match the CF location of one of the electrode probes. High frequencies were not tested because they would not affect drug delivery to the apex of the cochlea (i.e., the traveling waves stop near the CF location.) If the sound facilitation were most prominent near the peak of traveling waves, the low-frequency tone would be inefficient in facilitating drug delivery

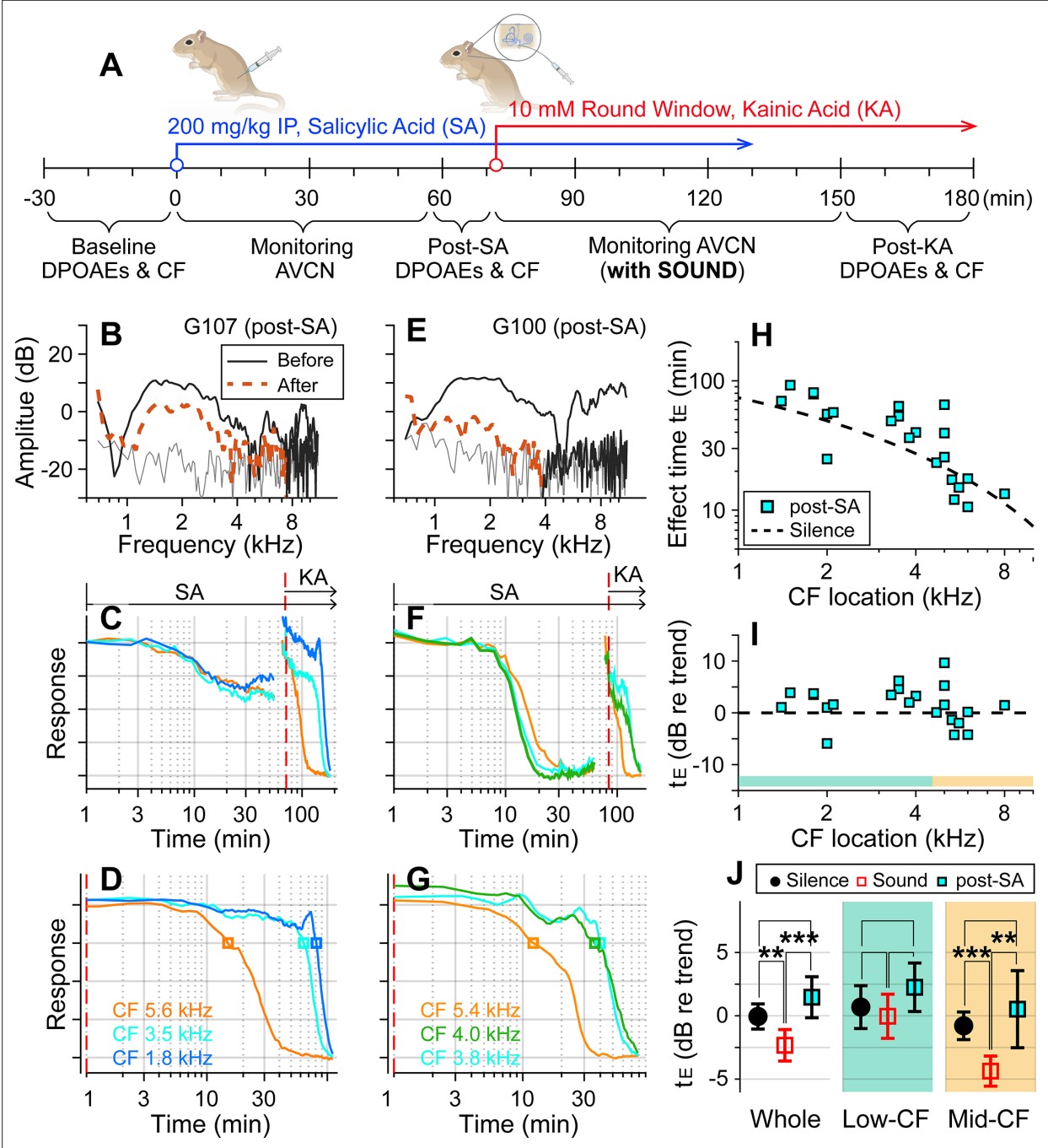

**Figure 2.** Suppressing motility of outer hair cells by salicylate. (**A**) Under the 'post-SA' protocol, salicylic acid (SA, 200 mg/kg) was administered IP before applying kainic acid (KA) at the round window. Broadband noise at 75 dB sound pressure level (SPL) was played during experiments. (**B**) Distortion product otoacoustic emissions (DPOAEs) before and after SA and KA delivery. (**C**) Overall response curves. IP SA was administered at t = 0. The red vertical line indicates when KA was applied to the round window. The arrows indicate the application spans of SA and KA. (**D**) Normalized neural responses after kainic acid application (the curves after the red line in **B**). (**E–G**) Another example of the post-SA protocol. (**H**) Effect time versus CF location of post-SA measurements (n = 22 from eight animals). The silence-case trend line is the same as *Figure 1*. (**I**) The effect time in dB with respect to the trend line. (**J**) Two-tailed *t*-tests between the effect times of sound, silence, and post-SA cases for the entire CF range (whole), low-CF locations (<4.5 kHz, n = 12 for the post-SA case), and high-CF locations (>4.5 kHz, n = 10 for the post-SA case).

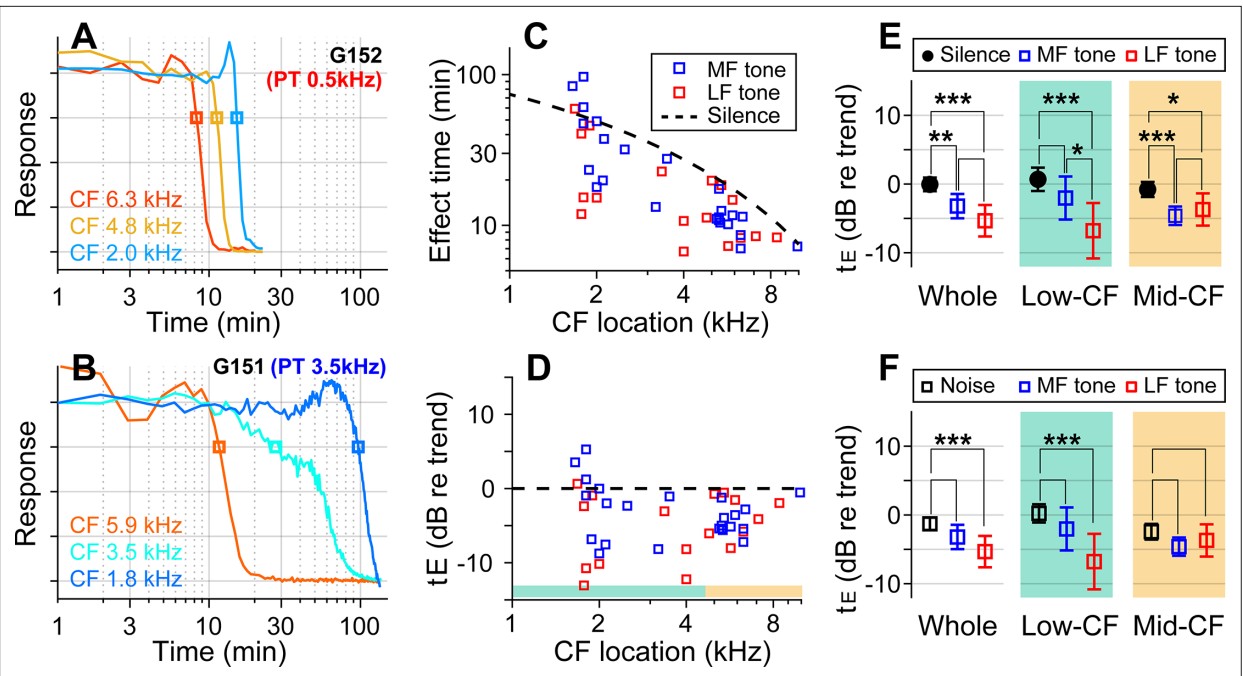

**Figure 3.** Pure tones yield shorter effect times than broadband noise. (**A**) Drug delivery during presentation of a tone at low-frequency (0.5 kHz 80 dB sound pressure level [SPL]) or (**B**) mid-frequency (3.5 kHz 80 dB SPL). (**C**) Effect time of drug delivery under the low- and mid-frequency tones. The trendline of the silence case is the same as *Figure 1I*. n = 18 from six animals for LF tone. n = 22 from eight animals for MF tone. (**D**) Normalized effect time. (**E**) Statistical comparison between the three cases (silence, low-, and mid-frequency-tone protocols). The two-tailed *t*-tests were performed on all CF locations, low-CF (<4.5 kHz, n = 12, 9 for MF- and LF-tone), and high-CF (>4.5 kHz, n = 10, 8 for MF- and LF-tone) locations. (**F**) Normalized effect time under three sound conditions compared: broadband noise, mid-frequency tone, and low-frequency-tone.

to high- to mid-CF locations. Whereas, if the phase velocity were crucial for fluid agitation, the low-frequency sound should be advantageous because of its long tail region with high phase velocity. If the traveling wave pattern set the cochlear fluid flow, the sound-facilitation effect could stall at one location, similar to the traveling waves.

With the pure-tone protocol, 45 data points were acquired from 15 gerbils. In two example cases, the effect time at low-CF locations (CFs near 2 kHz) was 15 min for the case of the 0.5 kHz tone (*Figure 3A*) and 100 min for the 3.5 kHz tone (*Figure 3B*). In the mid-CF locations, the effect time did not differ for different tone frequencies: the effect time ranged between 8 and 12 min for CF locations between 5.9 and 6.3 kHz (red curves of *Figure 3A and B*). In some cases, especially toward the low-CF locations, the response curve had bimodal decay patterns (e.g., the cyan curve in *Figure 3B*), resulting in a broader distribution in the effect time. The overall plots of effect time versus location (*Figure 3C and D*) show that most data points are below the trend line of the control (silence) case. That is, the pure-tone stimulation facilitated the delivery throughout the cochlea. According to the statistical analysis of effect time differences between the silence and the pure-tone protocols (*Figure 3E*), low-frequency tones facilitated the drug delivery for both low- (p<0.001) and mid-CF locations (p=0.011). Mid-frequency tones facilitated for mid-CF locations (p<0.001) but not significantly for low-CF locations (p=0.084). Low-frequency tones were more effective than broadband noise especially at the apical low-CF locations (*Figure 3F*, p<0.0005 for all CF locations, p<0.00001 for apical locations with CF <4.5 kHz).

*Figure 4* summarizes our findings for all four cases—control (delivery in silence, *Figure 4A and B*), sound (broadband noise at 60 or 75 dB SPL, *Figure 4C and D*), post-SA sound (SA applied intraperitoneally [IP] or to the round window 30–40 min before kainic acid application, broadband noise at 75 dB SPL, *Figure 4E and F*), and pure-tone sound (either 0.5 kHz or 3–6 kHz at 80 dB SPL, *Figure 4G and H*). The broken trend lines shown in all panels are the same, obtained by a curve fit to the control (silence) data using the 1D diffusion equation. Broadband sounds decreased the effect time in the mid-CF locations (CF > 4.5 kHz) but had little effect in the low-CF locations (CF < 4.5 kHz, *Figure 4C and D*). While measurements at two levels (60 and 75 dB SPL) were pooled in *Figure 1*, they are

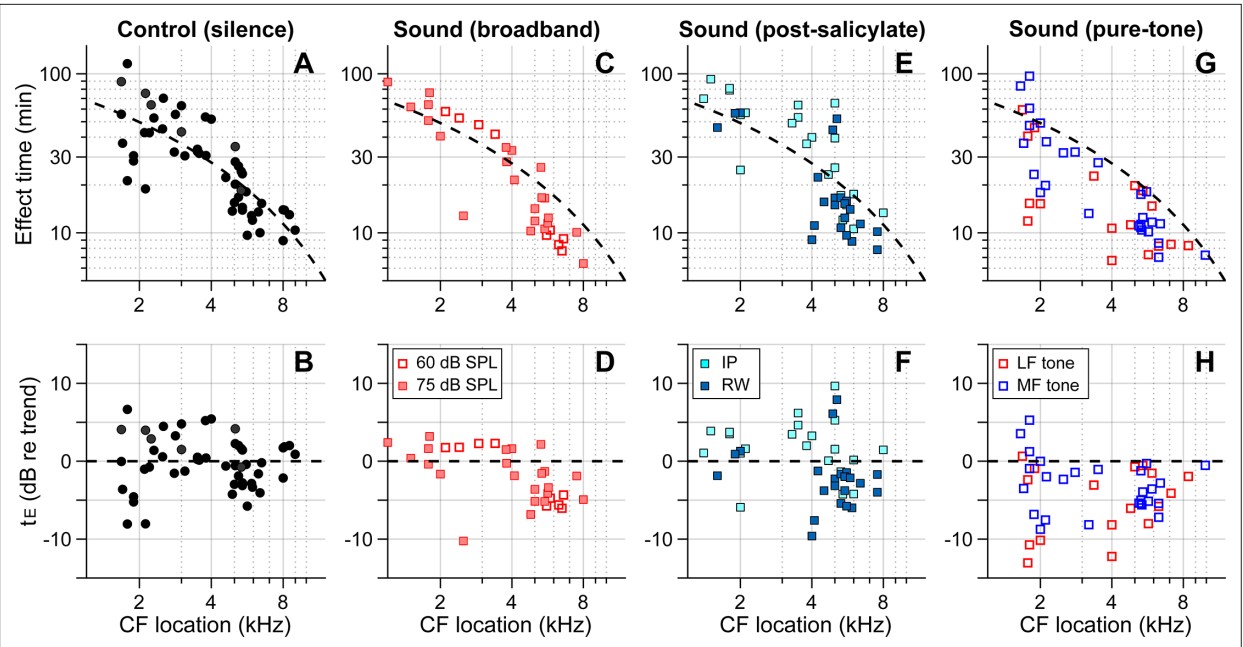

**Figure 4.** Outer-hair-cell motility facilitates inner-ear drug delivery. Summary of the effects of different acoustic and physiological conditions on inner-ear drug delivery. The y-axes represent absolute effect time in minutes (top) and relative effect time in dB w.r.t the trend line (bottom). The same trend line (broken lines, obtained from the control case) is presented for all panels. (**A, B**) Effect time in silence. (**C, D**) Effect time under 60 or 75 dB sound pressure level (SPL) broadband (0.1–12 kHz) sounds. (**E, F**) Effect time when outer-hair-cell motility was suppressed by salicylate. Before kainic acid application, the salicylic acid solution was administered systemically (intraperitoneally [IP], 200 mg/kg) or locally (round window [RW], 10 mM). (**G, H**) Effect time under 80 dB SPL pure tones. A 0.5 kHz (low-frequency [LF]) or 3–6 kHz (mid-frequency [MF]) tone was presented during drug delivery. The mid-frequency tone was chosen to match one of the probe frequencies.

distinguished in *Figure 4C and D*. Although the mean effect time of the 75 dB SPL data was smaller than for 60 dB SPL in the mid-CF region, supporting the sound-facilitation hypothesis, a larger sample number will be required to confirm the effect of sound level. The sound effect disappeared after inhibiting outer-hair-cell motility with salicylate (*Figure 4E and F*). In *Figure 2*, only IP-administered data of the post-SA case are presented. *Figure 4E and F* include additional data on the local (round window) application of salicylate before applying a solution containing 10 mM kainic acid and 100 mM salicylic acid. While the local delivery case reduced the mean effect time in the mid-CF region, the effect times of both post-SA cases were not significantly different from the control case. For the mid-CF region, the difference in mean effect time between the two post-SA cases was ascribed to the incomplete inhibition of outer hair cells in the local delivery case. The most prominent effect was observed when the drug was delivered while playing a low-frequency tone. For instance, 17 out of 18 effect times for the LF-tone case (red square symbols, *Figure 4G and H*) were below the trend line of the control (silence) case.

The mechanism of sound facilitation for drug delivery was investigated with computer model simulations. Our computer model is distinguished from most cochlear mechanics models in three aspects. First, it incorporates the interactions between the Corti fluid and OoC structures. Second, the nonlinear advection term in the Navier–Stokes equation was not neglected. Finally, both diffusion and fluid advection were considered. Our simulated results support the hypothesis that the cross-sectional area change of the OoC performs like a peristaltic pump (i.e., active outer hair cells are the fluid pumps) (*Shokrian et al., 2020*). *Figure 5* shows half of the cochlear fluid space—the scala tympani and the OoC fluid spaces (omitting the scala media and scala vestibuli). We simulated two cases: in the silence case, the drug delivery was purely governed by diffusion (*Figure 5A*). In the LF-tone case, both diffusive and advective mass transports contributed (*Figure 5B–D*). For the LF-tone case, a 1 kHz tone at 80 dB SPL sound was played continuously. The contours in *Figure 5A and B* represent the concentration of the substance at 30 min after the onset of delivery. As expected from Fick's law and the log-scale plots of *Figure 4*, the gradient over distance decays logarithmically. That is, contour lines in *Figure 5A* have similar spacings in the log scale. However, such a logarithmic gradient was apparently

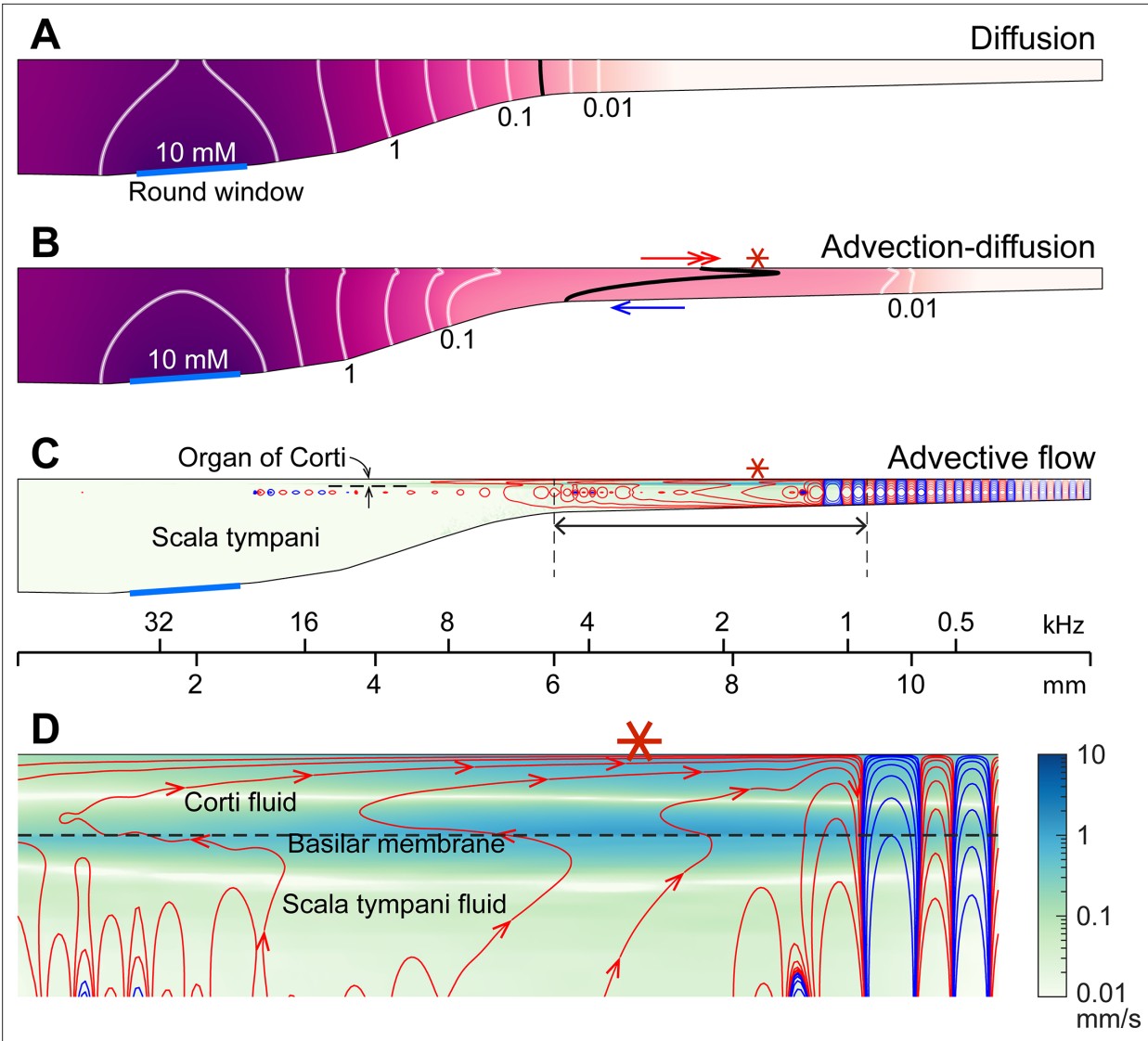

**Figure 5.** Active outer hair cells drive fluid flow. Inner-ear drug delivery was simulated using a computational model. (**A**) Drug delivery due to diffusion. The concentration profile over the cochlear length at 30 min after drug application to the round window. (**B**) Drug delivery with 80 dB sound pressure level (SPL), 1 kHz sound. The black contour line demonstrates how advection affects drug delivery. (**C**) Steady-state drift flow of the cochlear fluids. Red and blue curves represent the clockwise and counterclockwise streamlines. (**D**) The region with the fastest advection. The drift velocity (green color contour) is as large as a few mm/s.

disturbed between the 0.1 and 0.01 mM contour lines in *Figure 5B*. The contour line is stretched toward the right (red double-headed arrow) in the top Corti fluid layer and to the left (blue arrow) in the bottom scala fluid layer. This stretched contour line was due to the advective flow. In *Figure 5C and D*, the contour lines represent the streamlines (i.e., particles would drift along the contour lines). The red and blue colors indicate clockwise and counterclockwise circulation, respectively. *Figure 5D* presents the fluid streaming near the peak of the traveling wave (6–9.5 mm from the base, Corti fluid and top 120-μm-thick scala fluid). The maximum streaming velocity is as substantial as a few mm/s. Our simulation shows that active outer hair cells generate a strong stream toward the apex along the tunnel of Corti. The displaced fluid in the apex flows toward the base along the bottom of the scala tympani. The strong advection takes place in the tail of the traveling wave, over a few mm.

# Discussion

We showed that sound exposure increases drug delivery from the round window to apical cochlear locations. Sound-facilitated drug delivery was compromised when outer-hair-cell motility was inhibited. Low-frequency tones were more effective than broadband noise in delivering substances to the apex of the cochlea. These results not only highlight new strategies for improving drug delivery throughout the cochlear spiral, but reveal a new function of outer hair cells, beyond sound amplification.

Neural responses near CF are amplified, showing enhanced tuning. In contrast, outer hair cells boost the vibrations in the OoC non-selectively (*Cooper et al., 2018*; *He et al., 2018*; *Fallah et al., 2019*). To reconcile the nonselective input (untuned OoC vibrations) and the selective output (tuned neural signals), previous studies suggest that outer hair cells accumulate their power along cochlear traveling waves, referred to as global amplification (*Zagadou and Mountain, 2012*; *He et al., 2022*). However, experimental evidence is incongruent with the global amplification theory (*Fisher et al., 2012*; *Versteegh and van der Heijden, 2013*; *Dewey et al., 2019*). According to the two-tone suppression experiment of *Dewey et al., 2019*, the suppression of outer hair cells far basal from the CF location does not affect the response at the CF location. Available evidence shows that the amplification due to outer-hair-cell action occurs between a location one half-octave basal to the peak and the peak location (*Dong and Olson, 2013*; *Fallah et al., 2019*). The present study provides new insights into the nonselective outer hair cell action.

Broadband action of outer hair cells helps cochlear-fluid circulation. Our pure-tone experiments (*Figure 3*) and simulation (*Figure 5*) show that the region of sound-facilitated drug delivery spanned regions far basal to the CF location of a given sound frequency. Such sound facilitation disappeared when outer-hair-cell motility was suppressed (*Figure 2*), which implied that active outer hair cells are the stirrers in the cochlea that drive lymphatic-fluid circulation. From an evolutionary perspective, mammals stretched their hearing organ to encode broader bandwidth with good frequency resolution. The longer cochlea might have entailed a challenge: maintaining the freshness of the stationary lymph along a long and narrow tube is nontrivial. For instance, a local failure in homeostasis could cascade along the cochlea, making the system vulnerable to disturbance or trauma. The OoC in the basal cochlea suffers from a maintenance burden: a greater influx of potassium (greater MET current) and a smaller clearance surface. We hypothesize that the longitudinal circulation of lymphatic fluid serves as a protective mechanism for the fluid homeostasis of the mammalian cochlea. One potential implication is that an acoustically rich environment could be beneficial in maintaining healthy hearing as well as recovering from damaged hearing.

Our study complements existing studies regarding cochlear fluid homeostasis and differs from previous studies in several ways. The intrastrial fluids (extracellular fluids in the stria vascularis) have been more thoroughly investigated because the three layers in the stria vascularis (marginal, intermediate, and basal cells) maintain the endocochlear potential (*Wangemann, 2006*). Equilibrium in the Corti fluid has been sparsely investigated because its electrochemical gradient is modest compared to that of the intrastrial fluids (*Johnstone et al., 1989*; *Zidanic and Brownell, 1990*). Local electrochemical balance in the cochlear fluids has been considered within a radial section (*Quraishi and Raphael, 2008*; *Patuzzi, 2011*; *Nin et al., 2012*). Our study is focused on the longitudinal (global) equilibrium along the cochlear coil and did not consider the equilibrium across the stria vascularis cell layers. To examine whether the longitudinal fluid flow driven by outer hair cells is strong enough to affect cochlear fluid homeostasis, future studies should measure the $K^+$ equilibrium and recycling along the length of the Corti fluid under sound and silence conditions.

The nature of the streaming pattern supports the feasibility of targeted inner-ear drug delivery. Two characteristics implicated in the pure-tone experiments (*Figure 3*) became apparent in the model simulations (*Figure 5*): the advection caused by active outer hair cells created a 'fast track' and a 'stalling point' for mass transport. For instance, the span between locations 5.5 and 9 mm is the fast track of mass transport (the arrows of concentration contours in *Figure 5B* and the clockwise advective flow in *Figure 5C*). In this fast track, the distance between contour lines is much larger than the other regions or the pure diffusion case. The contour line in the fast track (black curve, *Figure 5B*) is stretched toward the apex in and near the Corti fluid and toward the base at the bottom of the scala tympani, indicating longitudinal fluid circulation. In contrast to the fast-track region, there was a stalling point where the transport stopped. The stacking contours near 10 mm in *Figure 5B* are caused by the countering circulation loops at 9 mm in *Figure 5C*. Taking advantage of these 'fast

tracks' and 'stalling points', one could design a sequence of sounds to deliver a substance from the round window to a targeted location in the cochlea.

Further work is needed before our findings can be used to address practical problems. Although we showed that low-frequency tone is superior to a high-frequency tone or a broadband sound in facilitating inner-ear drug delivery, we have not identified the optimal delivery scheme. While the gerbil cochlea is a better model to study longitudinal mass transport compared to the shorter mouse cochlea (12 mm versus 6 mm), substantial extrapolation is inevitable to predict drug delivery in the 34-mm-long human cochlea out of the present results. Extensive experiments would be needed to examine different SPLs at different frequencies and their combinations in a longer cochlea. Guidance from reliable theoretical models will help to design more efficient experiments and to interpret measured data more rigorously. Our present computer model incorporated the nonlinearity in advection-diffusion dynamics but did not implement the compressive nonlinearity of the sensitive cochlea. To analyze cochlear mass transport under stimulation by complex sounds, a model must address a long list of physical and physiological problems: deformable OoC, Corti fluid dynamics, nonlinear hair cell mechano-transduction, outer-hair-cell motility, permeable basilar membrane, nonlinear fluid dynamics including advection, diffusion solved simultaneously with advection, and the electrochemical equilibrium across the OoC. Our present model includes the essential components of these, but some are extensively simplified (e.g., outer-hair-cell motility, basilar membrane permeability), and some are absent (e.g., hair-cell nonlinearity, electrochemical equilibrium). Fortunately, there are other theoretical models that represent these aspects of cochlear physics and physiology that are lacking in our present study.

## Materials and methods
### Experimental approach
#### Animal preparation
Sixty-four Mongolian gerbils (*Meriones unguiculatus,* 25 males and 39 females, Charles River Laboratories) of 2–3 months of age were used in the study. All experimental procedures were in accordance with NIH guidelines and approved by the University of Rochester Animal Care and Use Committee (protocol # 101066). Animals were anesthetized with an IP injection of ketamine (140 mg/kg) and xylazine (6 mg/kg) and placed on a closed-loop heating pad to maintain body temperature at 37 ± 0.5°C. Ketamine and xylazine were then administered continuously through a subcutaneous catheter at a baseline rate of 85 and 3.5 mg/kg/hr, respectively, and adjusted as needed to maintain an areflexic state. Supplemental IP injections of ketamine and xylazine were occasionally administered as required. A tracheotomy and intubation ensured an open airway. Breathing rate was monitored using a thermistor placed in front of the trach tube. The right pinna was removed, and the bulla was exposed. A hole made in the dorsal and caudal bulla allowed access to the temporal bone and round window. After securing the skull using dental acrylic, customized headpost and ear bars, and a stereotaxic frame (Kopf Instruments, Model 900), a craniotomy was made in the temporal bone for electrode penetration through the dura to the AVCN (for details, *Frisina et al., 1982*). The head was oriented with the nose pitching downward at 45° so that the round window niche faced upward to better contain the pharmaceutical solution.

#### Drugs used
Pharmacological agents included the kainic receptor agonist kainic acid (Abcam and Tocris) and sodium salicylate (Sigma-Aldrich). A 10 mM kainic acid solution (volume of 4 μL) was applied at intact round window niche. The kainic acid solution was prepared in Hank's balanced salt solution (HBSS, Sigma-Aldrich). No discernible effects of different solvents on the AVCN responses were observed (data not shown). Sodium salicylate was dissolved in HBSS for local round window delivery at a concentration of 100 mM (volume of 4 μL) or dissolved in saline for IP injection at a dose of 200 mg/kg (volume of 0.3 mL).

#### Measurement protocol
All experiments were performed in a soundproof chamber. A pair of speakers and a microphone (ER-2 and ER-10, Etymotic) were plugged and sealed into the ear canals through a coupler. The SPL in the

ear canal was calibrated between 0.1 and 16 kHz. A reference electrode was inserted into the muscles of the back of the neck. A multichannel electrode (Q1x4-10mm-100-177-EIB, Neuronexus or ASSY-37 Fb Acute 32 channel, Cambridge Neurotech) was advanced 2–3 mm into the AVCN using a hydraulic microdrive (Model 640, Kopf Instruments), with a rostro-caudal angle between 60 and 65° and lateral angle between 40 and 50° with respect to the interaural line. The CFs of the four multiunit channels were identified using tones. Typically, three channels were identified showing well-defined frequency tuning and distinct CFs (*Figure 1A and D*). Neural responses were amplified and bandpass filtered between 0.75 and 10 kHz, then sampled (National Instruments PCIe-6251, 50 kHz or Intan RHD 2000, 30 kHz) for storage on a computer hard drive. Neural recordings were transformed with a Teager energy operator to enhance spiking activity (*Choi et al., 2006*), and the average value of the Teager-transformed response waveform was taken as a measure of the neural response amplitude. Typical time courses of neural signals are shown in *Figure 1C and F* (kainic acid was administered at t = 0). Experiments lasted between 1 and 3 hr.

Drug solutions were applied either at the intact round window niche using a 100 µL Hamilton syringe or through IP injection. After experiments, it was confirmed that the solution was still contained in the round window niche. While kainic acid diffused into the cochlea, the neural signals from the AVCN were recorded under three acoustical conditions: for the 'Sound' protocol, 1.1 s noise pips (60 or 75 dB SPL, 0.1–12 kHz bandwidth, 0.8 s duration including 0.15 s onset/offset ramps) were presented continually. After 48 noise pips, one 1.1 s silent pause and three CF tone pips followed (a total of 51 pips and a pause make a 57.2 s sequence). The CF tone pips were presented at the level of 35 dB SPL to monitor neural responses. The silence pause was to monitor spontaneous neural responses. The sequence was repeated until neural signals at the lowest CF site were completely abolished. The neural responses presented in this study are the 'driven responses' obtained by subtracting the spontaneous responses from the responses to the 35 dB CF tones. For the 'Silence' or 'Pure-tone' protocol, the noise pips of the Sound protocol were replaced with either silence pauses or a pure tone at 80 dB SPL. The pure tone frequency was either 0.5 kHz or a frequency between 3 and 6 kHz matching one of the mid-frequency probe channels. Stimuli were generated in MATLAB (50 kHz sampling frequency), converted to analog (National Instruments PCIe-6251), adjusted to the desired level using analog attenuation (PA5; Tucker Davis Technologies, Alachua, FL), and presented using a headphone buffer amplifier (HB7; Tucker Davis Technologies).

## Distortion product otoacoustic emissions (DPOAEs)

DPOAEs before and after the kainic acid delivery measurement were used to confirm that outer hair cell function did not change (*Figure 1C and F*). DPOAEs were recorded in response to swept primary tones, $f_1$ and $f_2$, with $f_2$ increasing linearly from 0.5 to 10 kHz over 4 s, and $f_2 = 1.25 f_1$. Primary tones (4.05 s duration including 25 ms raised-cosine on/off ramps, with 300 ms inter-stimulus intervals) were presented from separate earphones with equal sound levels, from 40 to 70 dB in 10 dB steps. DPOAEs were amplified by 40 dB prior to sampling and were averaged across 10 stimulus presentations. DPOAE level at $2f_1 - f_2$ was estimated using a least-squares fitting procedure (*Long et al., 2008*) in MATLAB, based on a 100 ms Hann window shifted in 20 ms steps (*Wong et al., 2019*). The same analysis was applied to a 'null' response, computed as the difference between average even and odd-numbered stimulus presentations, to determine the noise floor.

## Modeling approach

Cochlear fluid dynamics were divided into three sequential stages. Different assumptions were used in each stage to represent the relevant physics appropriately and to make the computational analyses affordable. First, OoC vibration due to sound stimulation was estimated by solving fluid–structure interactions between scala fluids, the Corti fluid (extracellular fluid in the OoC) and the OoC. The active feedback from outer-hair-cell motility was incorporated into OoC mechanics. Second, the drift velocity of the fluids in the OoC and the scala tympani in response to OoC vibrations was computed. Drift velocity originates from the nonlinear convective term in the governing equations for the incompressible flow (*Shokrian et al., 2020*). The spatial resolution required to evaluate the drift velocity made this stage computationally costly. Unlike the first stage, the advective component of fluid motion was incorporated. The OoC vibrations obtained at the first stage were given as kinematic boundary conditions in this second stage. The fluid motion within the OoC, as well as in the scala tympani, was

computed. Third, the spatiotemporal pattern of substance concentration due to advection and diffusion was obtained. The drift velocity field obtained in the second stage was the input for this third stage. The first and second stages were simulated using custom-written MATLAB programs, while the third stage was solved using commercially available software (COMSOL multiphysics).

## OoC vibrations

The OoC complex structures were discretized into 3-D finite elements reflecting anatomical details. For instance, each hair bundle, hair cell, Deiters cell, or pillar cell was represented by a fully deformable beam. The tectorial and basilar membranes were presented by a meshwork of beams. Their mechanical properties were determined either by using known values or by comparing them with existing mechanical measurements (*Nam and Fettiplace, 2010*; *Liu et al., 2017*; *Marnell et al., 2018*). Mechano-transduction kinetics of hair bundle and outer-hair-cell electromotility were incorporated with the OoC finite-element model (*Nam and Fettiplace, 2012*; *Nam, 2014*). The scala-fluid domain was discretized into linear quadrilateral elements. The scala-fluid pressure was determined by two vibrating boundaries—one is the oval window, and the other is the OoC complex. In turn, the OoC structures vibrate due to differential pressure across the OoC.

## Drift velocity of fluids

From the round window to the inner hair cell, kainic acid molecules travel (drift) across two fluid spaces separated by the basilar membrane: (1) the Corti fluid and (2) the scala-tympani fluid. The Corti fluid includes the fluid in the tunnel of Corti, Nuel's space, gaps between outer hair cells, and the outer tunnel. These sub-spaces are interconnected to one another by a few micrometer gaps. For simplicity, the 3-D fluid spaces were reduced to 2-D after reducing the radial dimension. The top boundary of the 2-D Corti fluid was modeled as a deforming wall while the bottom surface shared with scala tympani was implemented as a permeable boundary representing the basilar membrane. The OoC deformation was represented by the relative motion between the top and bottom surfaces of the Corti fluid. The pressure ($p$) and velocity ($u$) of the incompressible fluid were determined by the Navier–Stokes and continuity equations.

$$\frac{\partial u}{\partial t} + u \cdot \nabla u = -\frac{1}{\rho}\nabla p + \nu \nabla^2 u, \tag{1}$$

$$\nabla \cdot u = 0, \tag{2}$$

where $\nu$ and $\nu$ are the kinematic viscosity ($70 \times 10^{-6}$ m$^2$/s) and density (1 g/cm$^3$). Note that this viscosity value is a hundred times the value of the water after *Prodanovic et al., 2019*. The second term in *Equation 1* representing advection renders this equation nonlinear. This term has been neglected in cochlear mechanics studies because it hardly contributes to forming the cochlear traveling waves. However, for the purpose of this work (to investigate cochlear mass transport due to advection), this term had to be incorporated despite the increased computational cost. The basilar membrane was considered to have finite permeability using Darcy's law. The fluid velocity passing through the basilar membrane ($u_p$) is proportional to the transmembrane pressure difference ($\Delta p$), or

$$u_p = -K\Delta p. \tag{3}$$

The value of the permeability coefficient, $K$, of 1 $m/\left(sPa\right)$ was empirically determined so that the Corti-fluid pressure level was not excessive (less than a few Pa). Preliminary simulation results showed that drift velocities were not sensitive to the chosen value of $K$, even over an order of magnitude from the chosen value. Because of the basilar membrane's permeability, in general, the fluid velocity just above or below the basilar membrane ($u_f$) is different from the basilar membrane velocity ($u_m$) by $u_p$, or

$$u_f = u_m + u_p. \tag{4}$$

When periodic stimulation was applied, the resulting fluid motion was also periodic. Albeit much smaller than oscillatory motion, there existed nonperiodic fluid motion that made fluid particles drift over time. When fluid motion is integrated over a period, $T$, the oscillatory component cancels out, leaving only drift. The drift velocity $u_D$ at position $x_p$ and time $t + 0.5T$ was computed as

$$u_D\left(x_p, t + 0.5T\right) = \frac{1}{T} \int_t^{t+T} u_E\left(x_p, \tau\right) d\tau, \tag{5}$$

where $u_E$ is the Eulerian velocity obtained as a solution of the Navier–Stokes equations.

## Diffusion

Mass transport is governed by the advection-diffusion equation.

$$\frac{\partial C}{\partial t} = D\nabla^2 C - u_D \cdot \nabla C, \tag{6}$$

where $C$ is the concentration of the transported substance and $D$ is the diffusion coefficient. The whole fluid domain is impermeable (or $\partial C/\partial n = 0$, $n$ is the vector normal to the surface) except for the round window and the basilar membrane. At the round window, the concentration level was constant at 10 mM. The effect time at a specific location was defined as the time to reach 1% of the round window concentration (100 µM).

## Computation

The mesh for the fluid dynamics model (the first stage, *Equations 1 and 2*) consisted of approximately 25,000 second-order quadrilateral elements with the same degrees of freedom for both velocity and pressure. The grid size of the mesh was determined considering the boundary layer thickness of $\left(\nu/\omega\right)^{0.5}$, where $\omega$ was the highest frequency component of stimulation. The time integration was carried out using the second-order Crank–Nicolson method. The least-squares finite element approach was used with a reduced-order integration to ensure stability. The time-step size was 10 µs, and the problem was solved in the time domain for 5 ms (sufficient time to reach a periodical steady state). This time length was long enough to arrive at the periodic steady state. The mesh resolution for diffusion analysis was approximately 1 µm, resulting in about 225,000 first-order triangular elements. The initial time step was equal to the period of the stimulating frequency (1 ms) and was increased adaptively based on the convergence rate.

# Acknowledgements

The authors thank Jonathan Becker and Kris Abrams for their technical assistance. This study was supported by National Institute of Health of the United States (NIDCD R01 DC020150, PI: J-H Nam). The funder had no role in study design, data collection and interpretation, or the decision to submit the work for publication.

# Additional information

## Funding

| Funder | Grant reference number | Author |
|---|---|---|
| National Institute on Deafness and Other Communication Disorders | R01 DC020150 | Jong-Hoon Nam |

The funders had no role in study design, data collection and interpretation, or the decision to submit the work for publication.

## Author contributions

Choongheon Lee, Data curation, Formal analysis, Investigation, Visualization, Methodology, Writing – original draft, Writing – review and editing; Mohammad Shokrian, Formal analysis, Validation, Investigation, Visualization, Methodology, Writing – review and editing; Kenneth S Henry, Conceptualization, Data curation, Software, Formal analysis, Investigation, Methodology, Writing – review and editing; Laurel H Carney, Conceptualization, Resources, Investigation, Methodology, Writing – review and editing; J Christopher Holt, Investigation, Methodology, Writing – review and editing; Jong-Hoon

Nam, Conceptualization, Data curation, Formal analysis, Supervision, Funding acquisition, Validation, Investigation, Visualization, Writing – original draft, Project administration, Writing – review and editing

**Author ORCIDs**
Choongheon Lee http://orcid.org/0000-0002-3080-681X
Mohammad Shokrian http://orcid.org/0000-0003-0360-323X
Kenneth S Henry https://orcid.org/0000-0003-1364-318X
Laurel H Carney http://orcid.org/0000-0002-4729-5702
J Christopher Holt http://orcid.org/0000-0001-7908-083X
Jong-Hoon Nam https://orcid.org/0000-0002-7477-5453

**Ethics**
All experimental procedures were in accordance with NIH guidelines and approved by the University of Rochester Animal Care and Use Committee (UCAR Reference # 101066).

Reviewer #1 (Public review): https://doi.org/10.7554/eLife.101943.4.sa1
Reviewer #2 (Public review): https://doi.org/10.7554/eLife.101943.4.sa2
Reviewer #3 (Public review): https://doi.org/10.7554/eLife.101943.4.sa3
Author response https://doi.org/10.7554/eLife.101943.4.sa4

## Additional files

**Supplementary files**
MDAR checklist

**Data availability**
The data and the analysis codes used in this study are available at https://doi.org/10.6084/m9.figshare.27681999. The source codes to simulate Figure 5 are available at https://doi.org/10.6084/m9.figshare.27682281.

The following datasets were generated:

| Author(s) | Year | Dataset title | Dataset URL | Database and Identifier |
|---|---|---|---|---|
| Lee C, Shokrian M, Henry KS, Carney LH, Holt JC, Nam J-H | 2024 | Data and analysis codes (Lee, Shokrian, et al. 2024 eLife) | https://doi.org/10.6084/m9.figshare.27681999 | figshare, 10.6084/m9.figshare.27681999 |
| Lee C, Shokrian M, Henry KS, Carney LH, Holt JC, Nam J-H | 2024 | Source codes for Figure 5 (Lee, Shokrian, et al. 2024 eLife) | https://doi.org/10.6084/m9.figshare.27682281 | figshare, 10.6084/m9.figshare.27682281 |

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
