## [Editor Report · eLife Assessment]

Although others have proposed that OHC electromotility subserves cochlear amplification by acting as a ‘fluid pump’, and evidence for this has been found using electrical stimulation of excised cochleae, this **fundamental** study substantially advances our understanding of cochlear homeostasis. This is the first report to test the pumping effect in vivo and consider its implications for cochlear homeostasis and drug delivery. The manuscript provides **compelling** evidence for OHC-based fluid flow within the cochlea.

---

## [Referee Report · Reviewer #1 (Public review)]

Summary:

The authors test the "OHC-fluid-pump" hypothesis by assaying the rates of kainic acid dispersal both in quiet and in cochleae stimulated by sounds of different levels and spectral content. The main result is that sound (and thus, presumably, OHC contractions and expansions) result in faster transport along the duct. OHC involvement is corroborated using salicylate, which yielded results similar to silence. Especially interesting is the fact that some stimuli (e.g., tones) seem to provide better/faster pumping than others (e.g., noise), ostensibly due to the phase profile of the resulting cochlear traveling-wave response.

Strengths:

The experiments appear well controlled and the results are novel and interesting. Some elegant cochlear modeling that includes coupling between the organ of Corti and the surrounding fluid as well as advective flow supports the proposed mechanism.

The current limitations and future directions of the study, including possible experimental tests, extensions of the modeling work, and practical applications to drug delivery, are thoughtfully discussed.

---

## [Referee Report · Reviewer #2 (Public review)]

Although recent cochlear micromechanical measurements in living animals have shown that outer hair cells drive broadband vibration of the reticular lamina, the role of this vibration in cochlear fluid circulation remains unclear. The authors hypothesized that motile outer hair cells facilitate cochlear fluid circulation. To test this, they investigated the effects of acoustic stimuli and salicylate on kainic acid-induced changes in the cochlear nucleus activities. The results reveal that low-frequency tones accelerate the effect of kainic acid, while salicylate reduces the impact of acoustic stimuli, indicating that outer hair cells actively drive cochlear fluid circulation.

The major strengths of this study lie in its high significance and the synergistic use of both electrophysiological recording and computational modeling. Recent in vivo observations of the broadband reticular lamina vibration challenge the traditional view of frequency-specific cochlear amplification. Furthermore, there is currently no effective noninvasive method to deliver the drugs or genes to the cochlea. This study addresses these important questions by observing outer hair cells' roles in the cochlear transport of kainic acid. The author utilized a well-established electrophysiological method to produce valuable new data and a custom-developed computational model to enhanced the interpretation of their experimental results.

The authors successfully validated their hypothesis, showing through the experimental and modeling results that active outer hair cells enhance cochlear fluid circulation in the living cochlea.

These findings have significant implications for advancing our understanding of cochlear amplification and offer promising clinical applications for treating hearing loss by accelerating cochlear drug delivery.

---

## [Referee Report · Reviewer #3 (Public review)]

Summary:

This study reveals that sound exposure enhances drug delivery to the cochlea through the non-selective action of outer hair cells. The efficiency of sound-facilitated drug delivery is reduced when outer hair cell motility is inhibited. Additionally, low-frequency tones were found to be more effective than broadband noise for targeting substances to the cochlear apex. Computational model simulations support these findings.

Strengths:

The study provides compelling evidence that the broad action of outer hair cells is crucial for cochlear fluid circulation, offering a novel perspective on their function beyond frequency-selective amplification. Furthermore, these results could offer potential strategies for targeting and optimizing drug delivery throughout the cochlear spiral.

---

## [Author Response]

The following is the authors’ response to the previous reviews.

**Reviewer #1 (Public review):**
Summary:The authors test the "OHC-fluid-pump" hypothesis by assaying the rates of kainic acid dispersal both in quiet and in cochleae stimulated by sounds of different levels and spectral content. The main result is that sound (and thus, presumably, OHC contractions and expansions) result in faster transport along the duct. OHC involvement is corroborated using salicylate, which yielded results similar to silence. Especially interesting is the fact that some stimuli (e.g., tones) seem to provide better/faster pumping than others (e.g., noise), ostensibly due to the phase profile of the resulting cochlear traveling-wave response.Strengths:The experiments appear well controlled and the results are novel and interesting. Some elegant cochlear modeling that includes coupling between the organ of Corti and the surrounding fluid as well as advective flow supports the proposed mechanism.The current limitations and future directions of the study, including possible experimental tests, extensions of the modeling work, and practical applications to drug delivery, are thoughtfully discussed.Weaknesses:Although the authors provide compelling evidence that OHC motility can usefully pump fluid, their claim (last sentence of the Abstract) that wideband OHC motility (i.e., motility in the "tail" region of the traveling wave) evolved for the purposes of circulating fluid---rather then emerging, say, as a happy by-product of OHC motility that evolved for other reasons---seems too strong.

We adjusted our tone to be less assertive.

Our measurements and simulations coherently suggest that active outer hair cells in the tail region of cochlear traveling waves drive cochlear fluid circulation.

**Reviewer #2 (Public review):**
Although recent cochlear micromechanical measurements in living animals have shown that outer hair cells drive broadband vibration of the reticular lamina, the role of this vibration in cochlear fluid circulation remains unknown. The authors hypothesized that motile outer hair cells may facilitate cochlear fluid circulation. To test this hypothesis, they investigated the effects of acoustic stimuli and salicylate, an outer hair cell motility blocker, on kainic acid-induced changes in the cochlear nucleus activities. The results demonstrated that acoustic stimuli reduced the latency of the kainic acid effect, with low-frequency tones being more effective than broadband noise. Salicylate reduced the effect of acoustic stimuli on kainic acid-induced changes. The authors also developed a computational model to provide a physical framework for interpreting experimental results. Their combined experimental and simulated results indicate that broadband outer hair cell action serves to drive cochlear fluid circulation.The major strengths of this study lie in its high significance and the synergistic use of electrophysiological recording of the cochlear nucleus responses alongside computational modeling. Cochlear outer hair cells have long been believed to be responsible for the exceptional sensitivity, sharp tuning, and huge dynamic range of mammalian hearing. However, recent observations of the broadband reticular lamina vibration contradict widely accepted view of frequency-specific cochlear amplification. Furthermore, there is currently no effective noninvasive method to deliver the drugs or genes to the cochlea, a crucial need for treating sensorineural hearing loss, one of the most common auditory disorders. This study addresses these important questions by observing outer hair cells' roles in the cochlear transport of kainic acid. The well-established electrophysiological method used to record cochlear nucleus responses produced valuable new data, and the custom-developed developed computational model greatly enhanced the interpretation of the experimental results.The authors successfully tested their hypothesis, with both the experimental and modeling results supporting the conclusion that active outer hair cells can enhance cochlear fluid circulation in the living cochlea.The findings from this study can potentially be applied for treating sensorineural hearing loss and advance our understanding of how outer hair cells contribute to cochlear amplification and normal hearing.
**Reviewer #3 (Public review):**
Summary:This study reveals that sound exposure enhances drug delivery to the cochlea through the nonselective action of outer hair cells. The efficiency of sound-facilitated drug delivery is reduced when outer hair cell motility is inhibited. Additionally, low-frequency tones were found to be more effective than broadband noise for targeting substances to the cochlear apex. Computational model simulations support these findings.Strengths:The study provides compelling evidence that the broad action of outer hair cells is crucial for cochlear fluid circulation, offering a novel perspective on their function beyond frequency-selective amplification. Furthermore, these results could offer potential strategies for targeting and optimizing drug delivery throughout the cochlear spiral.Comments on revisions:Thank you for addressing the comments and concerns. The author has responded to all points thoroughly and clarified them well. However, please include the key points from the responses to the comments (Introduction (3), (5) and Results 5) into the manuscript. While the explanations in the response letter are reasonable, the current descriptions in the manuscript may limit the reader's understanding. Expanding on these points in the Introduction, Results, or Discussion sections would enhance clarity and comprehensiveness.

Introduction (3): As inner-ear fluid homeostasis is maintained locally, longitudinal electro-chemical gradients, including the endocochlear potential, may vary along the cochlear length (Schulte and Schmiedt 1992; Sadanaga and Morimitsu 1995; Hirose and Liberman 2003).

Introduction (5): We do not want to distract the readers from the primary message by discussing different drug delivery methods into the inner ear. This paper is regarding active outer hair cells’ new role as the title suggests. An extensive discussion of drug delivery can confuse the theme of this work.

Results (5): High frequencies were not tested because they would not affect drug delivery to the apex of the cochlea (i.e., the traveling waves stop near the CF location.)